# Genesis of Two Types of Carbonaceous Material Associated with Gold Mineralization in the Bumo Deposit, Hainan Province, South China

**Zhengpeng Ding [1,2], Teng Deng [1,2,*], Deru Xu [1,2,*], Zenghua Li [1,2], Shaohao Zou [1,2], Lirong Li [1,2], Ke Xu [1,2], Yan Hai [1,2] and Wen Ma [1,2]**

[1] State Key Laboratory of Nuclear Resources and Environment, East China University of Technology, Nanchang 330013, China; 201810709012@ecut.edu.cn (Z.D.); lizenghua@ecut.edu.cn (Z.L.); shaohaozou@ecut.edu.cn (S.Z.); 201800818001@ecut.edu.cn (L.L.); 201910818022@ecut.edu.cn (K.X.); 201910709015@ecut.edu.cn (Y.H.); 201910818017@ecut.edu.cn (W.M.)

[2] School of Earth Sciences, East China University of Technology, Nanchang 330013, China

\* Correspondence: dengteng2015@ecut.edu.cn (T.D.); xuderu@ecut.edu.cn (D.X.)

**Abstract:** Carbonaceous material (CM) is common in meta-sediments and is generally interpreted to be intimately associated with gold mineralization. For the Bumo deposit in Hainan Province, South China, CM is mainly hosted by greenschist facies—to amphibolite-facies metamophic rocks of the Paleo—to the Mesoproterozoic Baoban Group, and by auriferous veins which could be used as an important gold prospecting indicator. However, the genesis of CM and its relationship with gold mineralization are still unclear. From the field work and thin section observations two types of CM occur, i.e., layered and veinlet. The layered CM occurred in CM-bearing black shales, up to meters thick, and prevails in the deposit. More importantly, Au-bearing sulfides are commonly distributed along the boundary between the quartz veins and layered CM. In contrast, the veinlet CM, co-precipitated with native gold and sulfides, has the thickness of micro- to centi-meters, and these thin veins occur in quartz veins and hydrothermally altered rocks. In addition, layered CM has a stringy shape and laminate structure, while veinlet CM occurs as isometric particles based on the Scanning Electron Microscope (SEM) analysis. The Raman carbonaceous material geothermometer indicates that layered CM with a high maturity is formed at elevated temperatures of 400–550 °C, consistent with X-ray diffraction (XRD) analysis. In contrast, veinlet CM with a low maturity is formed at 200–350 °C and generally consistent with gold mineralization. In addition, layered CM has $\delta^{13}$C values ranging from −30 to −20%, demonstrating a biogenic origin. Consequently, it is interpreted that layered CM is formed by a pre-ore metamorphic event during Caledonian, and its reducing nature promotes gold precipitation via destabilization of aqueous Au complexes or facilitating sulfidation. Veinlet CM is of hydrothermal origin, and its precipitation modified the chemical conditions of ore fluids, leading to the destabilization of Au complexes, which therefore are favorable for mineralization.

**Keywords:** carbonaceous material; Raman geothermometer; Au; gold deposit; Hainan

## 1. Introduction

Carbonaceous material (CM) was reported to be found in many metasediment-hosted orogenic gold deposits, such as the Macraes deposit in New Zealand [1], Cosmo Howley in Australia [2], and Syama, Obuashi, and Inata deposits in west Africa [3]. Previous studies proposed that CM can be generated by metamorphic and hydrothermal activities [4]. The relationships between CM and gold mineralization have been proposed to be: (1) Pre-mineralization CM as a reducing agent to cause the precipitation of native gold and sulfides [5], (2) pre-mineralization CM absorbing Au complexes and

leached by syn-ore fluids [6–8], (3) pre-mineralization CM as a physically weak part to facilitate shear zones that localize ore veins [9], and (4) syn-mineralization CM co-precipitated with native gold and sulfides due to hydrothermal activities [10].

CM is widely distributed in the NE-trending Gezheng shear zone, the most important gold mineralization area in Hainan Province, South China [11]. CM-bearing rocks mainly include mylonite, compound gneiss and schists in Precambrian metasediments, and they are especially common in many gold deposits, such as the Bumo, Beiniu, Baoban, Fengshuishan, and Baolun deposits [12]. Previous research suggests that CM in the Baoban Group is produced by regional metamorphisms [11], and some of them are used for gold prospecting indicators. However, no studies have been conducted on the geological and geochemical characteristics of CM, so the genesis and association with gold mineralization are still not clear.

The Bumo deposit, with a gold reserve of about 13 t and an average grade of 9.97 g/t, is hosted by metasediments in the Paleo- to Mesoproterozoic Baoban Group. Among all the gold deposits along the Gezhen shear zone, the Bumo deposit is the one closest correlated with CM-bearing rocks. Major ore veins are located near the CM, and the gold grade in individual veins is strongly affected by the amount of CM [13,14]. However, the CM in the Bumo deposit has never been studied previously. CM characteristics in terms of structure, maturity, temperature, and isotopic composition are still not clear. Therefore, its contributions to auriferous veining are still not well understood. This paper uses an integrated analytical approach, combined field and petrographic work, SEM, laser Raman, XRD, and carbon isotope, to identify the two types of CM and their genesis and relationships with the gold mineralization in the Bumo deposit.

## 2. Regional Geology

The Hainan Province (Hainan Island), an epicontinental-type island in the north of the South China Sea, is situated at the conjunction part among the Pacific, Eurasian, and Indo-Australia plates (Figure 1A) [15,16]. The tectonic movements are divided into six stages with the ages of Jinningian, Caledonian, Hercynian, Indosinian, Yanshanian, and Himalayan time [17,18]. The tectonic events produced two major sets of structures, i.e., EW- and NE-trending faults. Among them, the EW-trending structures mainly include the Wangwu-Wenjiao, Changjiang-Qionghai, Jianfeng-Diaoluo, and Jiusuo-Lingshui faults, while the NE-trending structures, were represented by the Gezhen and Baisha shear zones (Figure 1B) [19,20].

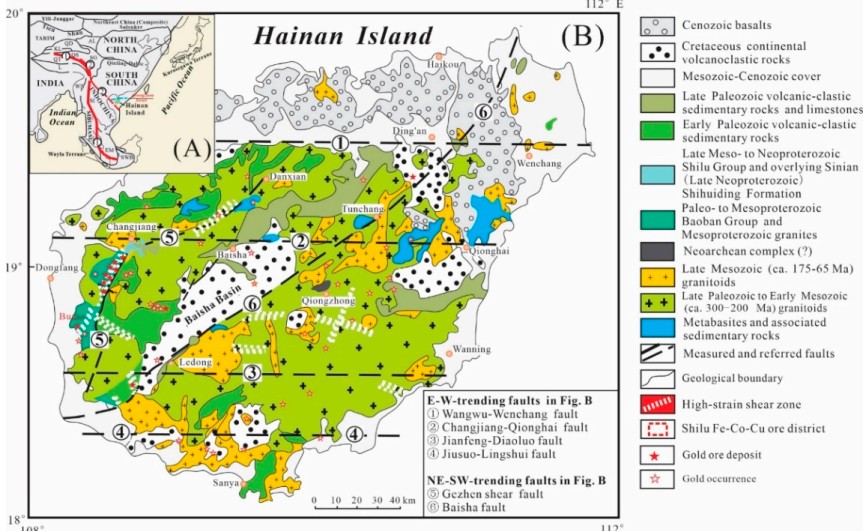

**Figure 1.** (**A**) Location map of Hainan Island (modified after Xu et al. 2017 [14]); (**B**) schematic diagram showing the main strata and magma units, gold deposits, and occurrences in the Hainan Island, South China (modified after Xu et al. 2017 [14]).

The Proterozoic strata in Hainan Province are well-developed (Figure 1B), including the Mesoproterozoic Baoban Group (ca. 1.8–1.45 Ga); [18], Meso-Neoproterozoic Shilu Group (ca. 1075–880 Ma); [21] and Neoproterozoic-Sinian Shihuiding Formation [14]. The Baoban Group predominantly distributed in the central Hainan Province, is composed of migmatitic gneisses, black shales, plagioclase-amphibole gneisses, and quartz-muscovite schists with amphibolite to granulite facies metamorphism [11,22]. The Shilu Group, distributed and located in the west Hainan, consists of a set of shallow sea-lagoon facies (iron-bearing) volcanic-clastic rocks, carbonaceous shales, and black shales. The contact between the Baoban Group and the overlying Shilu Group is hard to observe due to the cover of the Quaternary sediments [18,19]. The Shihuiding Formation consists of black slates, quartz sandstones, and quartzites intercalated with siltstones [16]. There is an unconformable contact between the Shilu Group and Shihuiding Formation [18,23]. The Precambrian rocks in Hainan Province likely went through extensive regional metamorphism during Caledonian [24].

It is noteworthy to mention that all the Precambrian metasediments are characterized with high contents of CM, up to volumetrically >10%, and it is hosted in the black shales, slates and carbonaceous shales, carbonaceous phyllites, and carbonaceous mudstones [14,18,19]. These carbonaceous rocks commonly occur over many tens of kilometers of rock band in the central Hainan Province [25]. Previous research indicates that CM is likely to be generated by metamorphism during the Caledonian period, and soured from Precambrian [26].

The exposed intrusive rocks account for about 40% of the land area, mostly located in the western-central Hainan Province. Most of those intrusions were produced at Yanshanian (175–65 Ma) and Hercynian-Indosinian (ca. 300–200 Ma) time [27], and the corresponding rocks are monzogranites-granodiorites and alkaline granites, respectively (Figure 1B). Late Paleozoic to early Mesozoic I-type granites are the most common and marked the beginning of the South China Indosinian Orogeny whereas the late Mesozoic calc-alkaline granites are considered as a response to the subduction of the Paleo-Pacific plate [28]. A few Mesozoic and Cenozoic basalts outcropped in several rift basins in northern Hainan Province [29].

More than 50 gold deposits and occurrences with a gold reserve of over 143 t have been discovered in Hainan Province, and most of them are located along the Gezhen ductile shear zone. The Gezhen ductile shear zone predominantly occurred as a narrow and long area, extending more than 150 km in the central-western Hainan Province [27]. The shear zone is characterized by multi-episodic tectonism, including earlier brittle-ductile and later ductile-brittle shearing deformation [21]. The major gold deposits in the Gezhen ductile shear zone are represented by the Bumo, Beiniu, Tuwaishan, Baoban, Niuling, and Erjia deposits [27]. CM-bearing rocks are important host rocks for native gold in Gezhen gold deposits, and the rocks mainly include carbonic phyllites, black shales, and carbonaceous slates [13].

## 3. Ore Deposit Geology

The Bumo gold deposit is situated in the southwestern part of the DongFang city, Hainan Province, and has an area of 56 km$^2$ [30]. The rocks in the Bumo deposit include the Baoban, Shilu Groups, and Quaternary sediments (Figure 2A). The Baoban Group is mainly composed of the quartz sericite schists, carbonaceous shales (CM-bearing rock), metamorphosed quartzose sandstones, phyllites, and migmatization schists. The Shilu Group consists mainly of metamorphosed quartzose sandstones, biotite adamellites, carbon slates (CM-bearing rock), and mylonite schists. The Quaternary loose sediments are mainly composed of unconsolidated sediments of sand [27]. CM-bearing rocks are quite common within the Baoban and Shilu Groups in the Bumo deposit, such as carbonaceous phyllites, carbonaceous mudstones, and shales [14]. Locally, CM was aggregated along faults to form a CM layer with the thickness up to one meter, and they are commonly associated with gold-quartz veins [14,20].

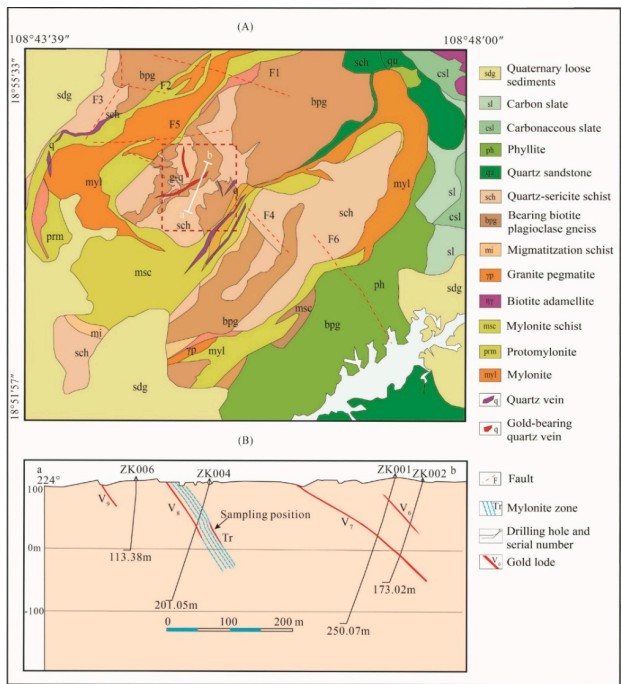

**Figure 2.** (**A**) Simplified geological map showing the location of the gold deposits in the Gezhen brittle-ductile shear zone, western Hainan and (**B**) cross section alone line a–b, modified after Xu et al. 2017 [14].

Magmatic rocks are not common in the mining area, they only have a few Hercynian and Indosinian period monzogranites with exposure to the southern side of the deposit [27]. Even though the possible existence of concealed intrusions is suggested by Controlled Source Audio-Frequency Magotelurics (CSAMT) and Audio-Frequency Magotelurics (AMT). There are some Yanshanian pegmatite, diorite porphyry, and diorite outcrop outside in the north of the deposit area [31]. There are three sets of faults in the deposits, i.e., the NE-, NW-, and EW-trending faults, which host gold-bearing quartz veins, and are parallel to the host rock of the Baoban Group [32,33]. The NW- and EW-trending faults cut through the NE-trending faults, suggesting that the NW- and EW-trending faults formed after the NE-trending faults [22].

More than 20 orebodies have been discovered in the Bumo deposit. Most orebodies are irregular lodes controlled by the fault. The orebodies generally occurr SW—trending fault and parallel to each other (Figure 2A). The No. 2 orebody, consisting of V6–V9 lodes in the center of the Bumo deposit, is the ore vein with the largest tonnage (Figure 2B). This orebody dips at 90–97° to the SW and has an average ore grade of 5 g/t gold. Two mineralization styles are recognized, i.e., disseminated and vein type ores [13]. The wall rock alteration mainly includes pyritization, silicification, and sericitization [12]. Pyrite and arsenopyrite are the two principal ore minerals (Figure 3A), and there are also some sphalerite, chalcopyrite, and galena (Figure 3B). The results of previous EMPA analyses showed that most arsenopyrite, pyrite, and sphalerite grains contain native gold [34]. The gangue minerals mainly include quartz, CM, calcite, and sericite (Figure 3C,D). Previous research indicates that the hydrothermal event can be divided into three stages [35], i.e., quartz (Q1)-pyrite, quartz (Q2)-sulfides, and quartz (Q3)-calcite stages, with the second stage as the main mineralization stage. Different from the white quartz (Q1) with euhedral structures (Figure 3E), Q2 grains are generally anhedral (Figure 3F) and its aggregates are commonly grey.

Two types of CM are recognized based on field work, hand specimen, and microscope observations, i.e., layered and veinlet CM. The layered CM occurred in the CM-bearing black shales of gold-bearing quartz veins, and the thickness can be up to one meter (Figure 4A,B). The mineral of layered CM generally shows a stringy shape (Figure 4C) and has a laminate structure (Figure 4D). More importantly,

sulfides generally precipitated along the boundary between quartz veins and layered CM (Figure 4E). Veinlet CM, crosscutting the early white quartz (Q1) grains (Figure 4F,G), constitute micro veins together with native gold and sulfides (Figure 4H). Under SEM, veinlet CM exhibits the shape of isometric particles (Figure 4I).

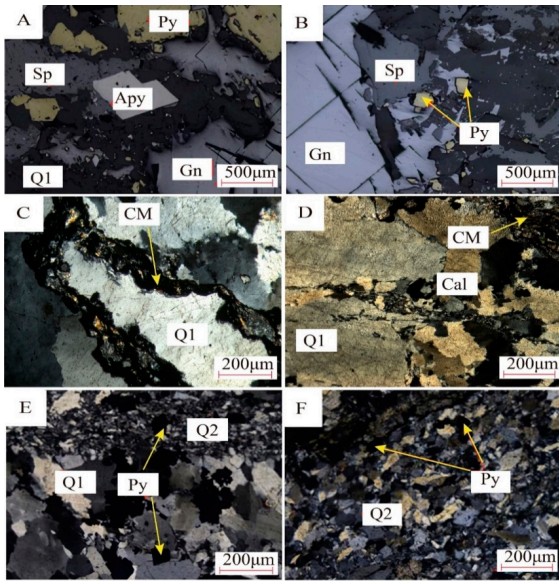

**Figure 3.** Photographs showing the mineral assemblages in the Bumo deposit. (**A**,**B**) Pyrite and arsenopyrite associated with galena, sphalerite, and white quartz (Q1; reflected-light); (**C**) carbonaceous material (CM) crosscut and replaced by greyish quartz (Q2) and sericite (cross-polarized light); (**D**) calcite crosscut white quartz (Q1; cross-polarized light); (**E**) white quartz (Q1) crosscut by greyish quartz (Q2) associated with pyrite (cross-polarized light); (**F**) greyish quartz (Q2) and pyrite (cross-polarized light). Apy: Arsenopyrite; Py: Pyrite; Sp: Sphalerite; Gn: Galena; Q: Quartz; Ser: Sericite; Cal: Calcite.

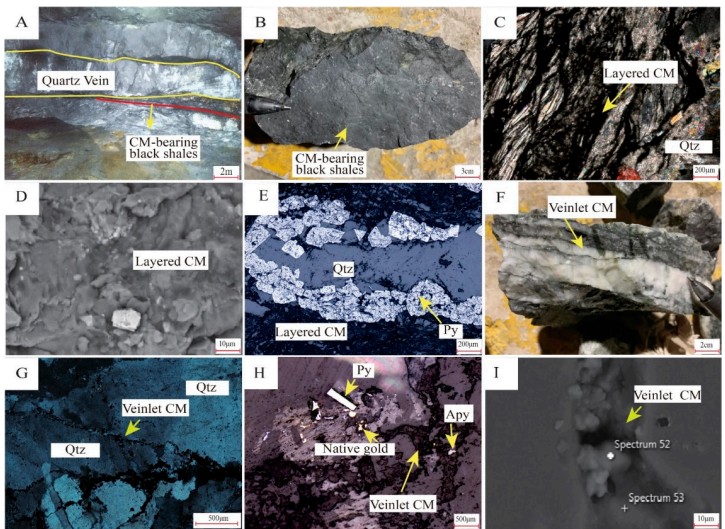

**Figure 4.** Photographs of the layered and veinlet CMs from the Bumo gold deposit. (**A**) Quartz vein and CM-bearing black shales coexisting in the ore body; (**B**) CM-bearing black shales; (**C**) filamentous particles of layered CM in quartz veins (cross-polarized light); (**D**) SEM image of layered CM; (**E**) layered CM with sulfides (all in black seams) formed in quartz veins (reflected-light); (**F**) detail of veinlet CM in quartz veins; (**G**) veinlet CM through the large coarse quartz particles (plane-polarized light); (**H**) bright point within the veinlet CM with Au and sulfides (reflected-light); (**I**) SEM image of veinlet CM. Qtz: Quartz; Apy: Arsenopyrite; Py: Pyrite.

## 4. Samples and Analytical Methods

A total of 24 samples were collected from the No. 2 ore body of Bumo deposit. CM thin sections were prepared from all samples. The layered CM was extracted from CM-bearing black shales samples, and they are milled to below 200 mesh powder. About 50 g of each sample was ground under ethanol for 5 min, such that the short time of pulverized would not affect the structure of layered CM. The chemical separation of layered CM samples was performed using a standard procedure using hydrochloric (HCl) and hydrofluoric (HF) acid treatment, thereafter heavy-liquid separation [36].

### 4.1. Petrography and Scanning Electron Microscopy

The petrographic analysis of the layered and veinlet CMs was used by a digital microscope (Leica DM4) and scanning electron microscopy (Nova Nano SEM450) at the State Key Laboratory of Nuclear Resources and Environment (East China University of Technology). Meanwhile, we coated 10–20 nm of gold in thin sections for SEM analysis that was because we want to be identified types of CM through the EDS system within SEM. The resolution is set at 1.0 nm and 15 kV. The resolution is set at 1.0 nm and light of 15 kV. The photos were taken by an electron beam exposure machine under the conditions of an accelerating voltage of 15 kV, filament current of 240 mA, and an electron beam diameter of 3.5 mm [37].

### 4.2. Raman Carbonaceous Material Geothermometer

Raman spectra were measured by the Renishaw inVia Reflex Raman spectrometer (Renishaw, UK) at the State Key Laboratory of Nuclear Resources and Environment (East China University of Technology) with a resolution of 0.5 cm$^{-1}$. The He–Ne ($\lambda$ = 633 nm) laser is used as the excitation source and is focused onto the sample with the Raman spectra of the cylindrical lens that has been recorded at normal temperature with perpendicular and parallel polarization of incident and scattered beams. Raman spectrum is a useful non-destructive tool for calculating the maturity of organic matter [38–40]. Advances in the instrument and data processing software have promoted increased applications of Raman spectroscopy, and this technique was simple, fast, and can be performed directly on petrographic thin sections or solid kerogen at present.

Specifically, the Raman spectrum has two bands, namely G (order) and D (disorder), which are related to the maturity of the nonrandom structure of the disorder domain in the organic matter. The D and G bands can demonstrate that the relevance with geological processes temperature are between 200 and 650 °C [38,39]. The G band is the only Raman vibration peak in pure graphite at about 1582 cm$^{-1}$ and is related to the in-plane vibration of the carbon atoms in the pure graphite. The D band which is located on 1350 cm$^{-1}$ becomes active in amorphous carbon, and its frequency depends on the laser beams. The result of a double resonance Raman scattering process was a confirmed source. Beyssac et al. (2002) [38] indicated that there is an additional band in the Raman spectrum of a coal mine: The D2 band was observed as a small peak in the G band peak at approximately 1620 cm$^{-1}$. These bands are ascribed to out-of-plane defects from tetrahedral carbons, or to a small crystallite size. With an increase of the graphitization degree, the G band becomes stronger than the D band and decides the degree of structural order increases with temperature, which is also the basis of the geothermal meter [38,39]. The degree of structural order in CM depends on two ratios: R1 (the specific value of disordered peaks height divided by the ordered peaks value) and R2 (the specific value of disordered peaks area divided by the ordered band area). Peak and area curve fitting and calculations of the Raman spectra were conducted by using the Peakfit 4.1.2 software. This can better reflect the difference between the peak value and area [38]:

$$R1 = \frac{D1}{G} \text{ (peak value of band)} \tag{1}$$

$$R2 = \frac{D1}{G + D1 + D2} \text{ (area of band)} \tag{2}$$

where indices Area and Height mean that the ratio base are on the band area and height. By using the R2 ratio, Beyssac et al. (2002) suggested that the R2 ratio is an important parameter in the metamorphic thermometer. They proved that the crystallinity of CM is strongly correlated with the peak metamorphic temperature and not concerned with the meta-morphic pressure. Their thermometer is designed according to the linear relationship between the metamorphic temperature and parameter R2:

$$T(°C) = -445R2 + 641 \tag{3}$$

On the other hand, Rahl et al. (2005) suggested a modified thermometer based on parameters R1 and R2, which is applicable in the temperature range of 100–650 °C with the fit parameter R2 more than 90%:

$$T(°C) = -737.3 + 320.9R1 - 1067R2 - 80.3638R1^2 \tag{4}$$

### 4.3. X-Ray Diffractometry (XRD)

The XRD patterns of layered CM was performed by the Bruker D8 Advance X-ray powder diffractometer with Cu K$\alpha$ radiation ($\lambda$ = 0.154 nm) at the State Key Laboratory of Nuclear Resources and Environment (East China University of Technology). The samples were set up on a low-background spinning specimen holder. In order to identify residual or newly formed mineral phases after hydrochloric (HCl) and hydrofluoric (HF) acid treatment, the XRD analysis in the range of 10–70° of 2θ were conducted until they were identified in all the layered CM samples. The step-scanned powder diffraction data was processed by the XFIT program [41], applying the split asymmetric Pearson profile shape function to yield the peak positions and full width at half maximum (FWHM) values. It is used for fitting the broad peak profile partially integrated into the background.

### 4.4. Carbon Stable Isotopes

For the measurement of layered CM samples, stable carbon isotopes were extracted and ~20 g of the powder was put it into pure orthophosphoric acid ($H_3PO_4$) at 75 °C for 3 h to extract the $CO_2$ gas. The evolved $CO_2$ gas was injected into a Finnigan MAT-253 stable isotope mass spectrometer and carrier helium gas. The China standard (GBW04405) has been calibrated using the NBS19 standard, which is used to calibrate $\delta^{13}C$ data relative to the V-PDB scale. Results are given in standard $\delta$-notation relative to the V-PDB international standard for $\delta^{13}C$ [42]. The analysis accuracy based on standard reproducibility was better than ±0.2% for $\delta^{13}C$ and measurements, respectively (1σ). These measurements were carried out using the Finnigan MAT-253 stable isotope mass spectrometer at the State Key Laboratory of Nuclear Resources and Environment (East China University of Technology).

## 5. Results

### 5.1. Raman Spectra of CM

There are two notable symbolic differences between the Raman bands of the layered and veinlet CMs. The layered CM has a lower D1 band peak than the G band, as well as an independent D2 band peak (Figure 5A). D1 and G bands of veinlet CM have similar peak heights and areas, and a D2 band appears only as a weak shoulder on the high wavenumber side of the G band (Figure 5B). We collected all of the layered and veinlet CMs. The Raman spectra data are shown in Table S1.

Previous research indicates that the graphitization of CM increases with rising temperatures, and the R2 ratios decreases significantly with rising temperatures [43,44]. In our study, the R2 values and temperatures are linearly correlated (Figure 6B). Meanwhile, the layered CM has lower R1 and R2 ratios than the veinlet CM (Table S1, Figure 6A–C). Two different calibration equations indicate similar temperature ranges. The Raman geothermometer indicates that the layered and veinlet CMs were formed at 400–550 °C and 200–350 °C, respectively (Figure 6D).

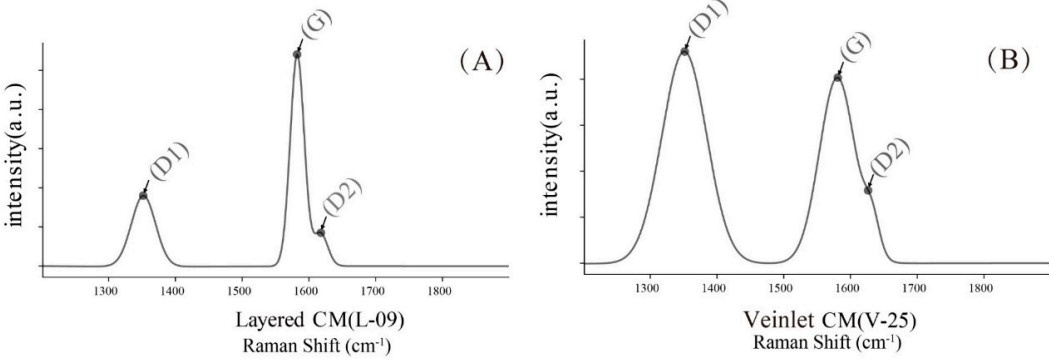

**Figure 5.** (**A**) Raman spectrum of layered CM; (**B**) Raman spectrum of veinlet CM.

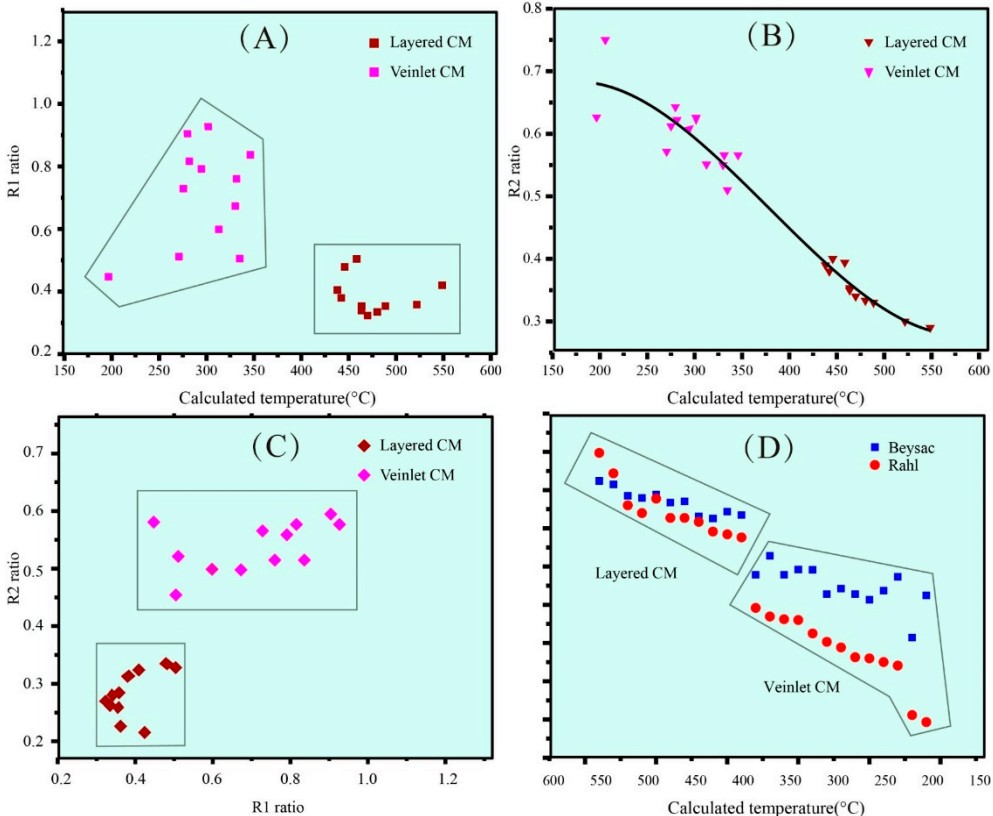

**Figure 6.** (**A**,**B**) R2 and R1 versus an independently estimated temperature. The different colors correspond to different types of CM. We demonstrate a linear correlation in R2 over the range 150 to 600 °C, but there is a little variation in R2 for temperatures over low temperatures. In contrast, R1 increases over the low-temperature range; (**C**) correlation between temperature and the Raman R1 and R2 ratios for layered and veinlet CM particles. The R1 and R2 ratios of CM particles from the Bumo deposit are given for comparison; (**D**) metamorphic temperature calculated from the Raman R2 value after Beyssac et al. (2002) [38] compared with metamorphic temperatures calculated from the Raman R1 and R2 values after Rahl et al. (2005) [39].

### 5.2. XRD Patterns of Layered CM

The X-ray diffraction analysis can be used to interpret spectrometric characteristics at different $d_{002}$ and $2\theta$ values, and determine the degree of graphitization [45]. Previous studies show that the $d_{002}$ values decrease with increasing graphitization. Figure 7A shows that the $d_{002}$ decreased continuously from 3.350 to 3.375 Å in layered CM samples, and all of them belong to the prehnite pumpellyite facies and amphibolite-facies. The presence of high crystallinity layered CM samples is indicative of their

formation under a high-temperature (400 °C-higher) environment. Furthermore, XRD patterns showed that layered CM samples were accompanied together with other minerals. Although, the 2θ values of 26.5° indicate the presence of CM and quartz (Figure 7B), we believe that the peak of 26.5° is CM instead of quartz based on our petrography (Figure 4B) and carbon stable isotope results (Figure 8).

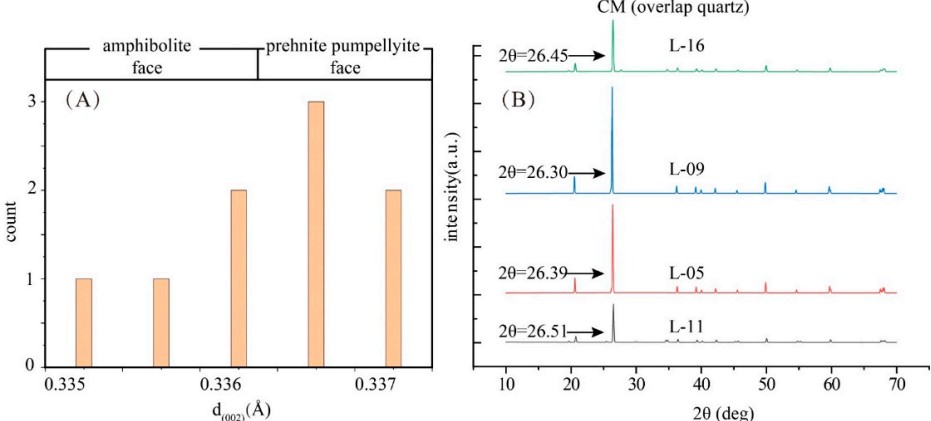

**Figure 7.** (**A**) XRD parameters of $d_{002}$ graphitic peak of layered CM from the Bumo deposit; (**B**) XRD patterns of the layered CM, accompanied by a high graphitic peak intensity at around 26.5° and other mineral peak intensity characteristics.

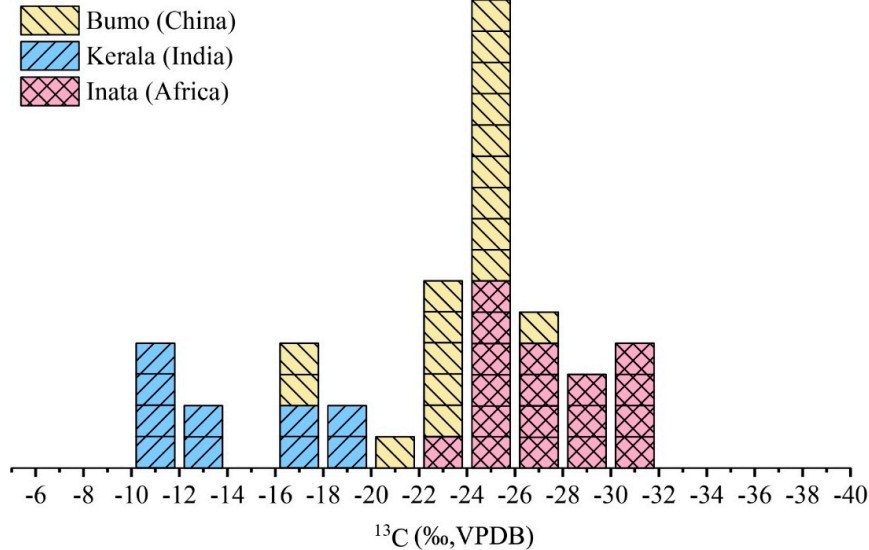

**Figure 8.** Frequency distribution of $\delta^{13}C$ values of layered CM from the Bumo deposit and compared with different gold deposits.

### 5.3. Stable Carbon Isotopic Analysis of Layered CM

Seventeen samples from the layered CM were analyzed for stable carbon isotopic analysis, and the $\delta^{13}C$ values are −20 to −27% (VPDB). As shown in Figure 8, all these data are plotted in the region of organic carbon. Previous research demonstrates that isotopic exchange kinetics in CM are sluggish, therefore subsequent hydrothermal fluids are not affected by the isotopic composition of CM [46].

## 6. Discussion

### 6.1. The Genesis of Layered and Veinlet CM

Many researchers indicate that CM can be formed by metamorphism of sedimentary rocks with high organic matter contents [47]. The organic matters initially contained in the sediments can be quite

different, such as lacustrine and marine algae [48]. Under the influence of metamorphism, organic matters can be decomposed into biopolymers, which can be later polycondensation reactions into geopolymers. CM was finally generated after the metasediments rich in geopolymers (such as black shales) went through brittle-ductile deformation in the shear zone [43], and when produced in this way generally occurs as thick layers up to several meters thick and commonly distributed in the footwall [49], and this CM commonly produced by metamorphism generally presents as black and grayish brown plate-like substances on hand specimens and flocculent body in microscopic images [50]. The generation commonly required the metamorphic grade with temperatures up to 400–550 °C [50].

The alternative genesis for CM is via the redox reaction between $CO_2$- and $CH_4$-bearing hydrothermal fluids in ore deposits. In the process of fluid infiltration, the geological process will drop the amount of $CO_2$ and $CH_4$ in fluids [51]. This CM generated by hydrothermal fluids is composed of tiny isometric particles with sizes lower than 5 mm and has a high disorder in SEM image. However, they also form veinlets locally along with quartz, carbonate, and ore-bearing sulfides. In addition, this CM associated with hydrothermalism is commonly formed instantly under low temperatures, so there is far less time for the CM to accumulate impurities, different from the CM which is produced by metamorphism, which develops over millions of years [9].

The layered CM commonly occurs in the CM-bearing black shales of fault in the Bumo deposit (Figure 4A), and present as flocculent textures under a microscope. The Raman carbonaceous material geothermometer indicates that the layered CM is formed at 450–550 °C (Figure 6D), consistent with the $d_{002}$ values of 3.35–3.375 obtained by XRD analysis (Figure 7). Previous fluid inclusion research suggests that the Bumo deposit was formed at 180–350 °C [52], a significantly lower temperature of layered CM. Meanwhile, the Baoban Group went through greenschist- to amphibolite-grade metamorphism during Caledonian [30] which reached up to 500 °C [14,18], similar to that of layered CM. In addition, $\delta^{13}C$ values of layered CM, ranging from −20 to −27%, demonstrated the origin of organic carbon [53]. Consequently, layered CM was interpreted to be generated by metamorphism of the Baoban Group during Caledonian, transferring the organic matter dispersed into graphite [22].

Different from layered CM, the veinlet CM generally occurs in ore-bearing quartz veins [54]. Veinlet CM grains with complete spherical structures in SEM image form micro-veins with native gold, arsenopyrite, and pyrite, crosscutting early large quartz grains. The Raman carbonaceous material geothermometer demonstrates that the veinlet CM was formed at 200–300 °C, consistent with the gold mineralization and significantly lower than the layered CM. Consequently, veinlet CM is likely to be of hydrothermal origin. Previous research indicates that the ore fluids of the gold deposits in the Gezhen shear zone are rich in $CO_2$ [30], and altered host rocks have high contents of hydrocarbon gas (such as $CH_4$, $H_2$, and $C_2H_4$). In addition, a lot of chlorite was discovered in the Bumo deposit. Most of the chlorite from the Bumo deposit belongs to the chamosite of which iron is mainly present as "Fe (II) complexes" [34]. Consequently, we can infer that the veinlet CM was interpreted to be formed by the following two redox reactions (L means ligand; e.g., $Cl^-$, $HS^-$) [55]:

$$CH_{4(g)} + CO_{2(g)} = 2C + 2H_2O_{(l)} \tag{5}$$

$$2FeL_n^{2-n} + 4H_2S_{(aq)} + CO_{2(aq)} \rightarrow 2FeS_2 + C_{(s)} + 2H_2O_{(l)} + 4H^+ + 2nL^- \tag{6}$$

### 6.2. Implication for Gold Mineralization

Two types of CM with multiple geneses are commonly discovered in many hydrothermal gold deposits [56]. CM with the metamorphic origin is an ideal reducing agent for Au complexes and sulfide precipitation. In addition, some scholars proposed that native gold were initially absorbed by CM grains in Precambrian metasediments, and was later leached by ore fluids [7]. These pre-ore CM are also physically weak, and are prone to faulting and provide space for ore fluid precipitation [9]. As for CM with the hydrothermal origin, its co-precipitation with sulfides altered the pH of ore fluids, led to the destabilization of Au complexes [57].

In the Bumo deposit, as a criterion for ore prospecting, layered CM is generally accompanied by gold-bearing ore veins. More importantly, Au-bearing sulfides are concentrated along the boundaries between the quartz veins and layered CM. Consequently, the spatial association between layered CM and Au-bearing sulfides in quartz veins does indicate that the gold deposition affected by a redox reaction with veinlet CM occurs as micro-veins together with sulfides and gold, and they are produced by reaction 6 [9,58,59]. Layered CM through chemical reaction 7 (L means ligand; e.g., $Cl^-$, $HS^-$) [5]:

$$2AuL_n^{1-n} + C_{(s)} + 2H_2O_{(l)} \rightarrow 2Au_{(s)} + CO_{2(aq)} + 4H^+ + 2nL^- \tag{7}$$

Based on the above description, the mineralization model for the Bumo gold deposit is outlined as: (1) Layered CM was formed by the pre-ore Caledonian metamorphism at the temperatures of 450–550 °C, and this CM is physically weak, leading to stress concentration and shear deformation; (2) the reducing environment and space provided by layered CM promoted the ore mineral precipitation (Figure 9A,B); (3) veinlet CM is also produced by the water-rock interaction at 200–350 °C, coeval with sulfides, and native gold (Figure 9B).

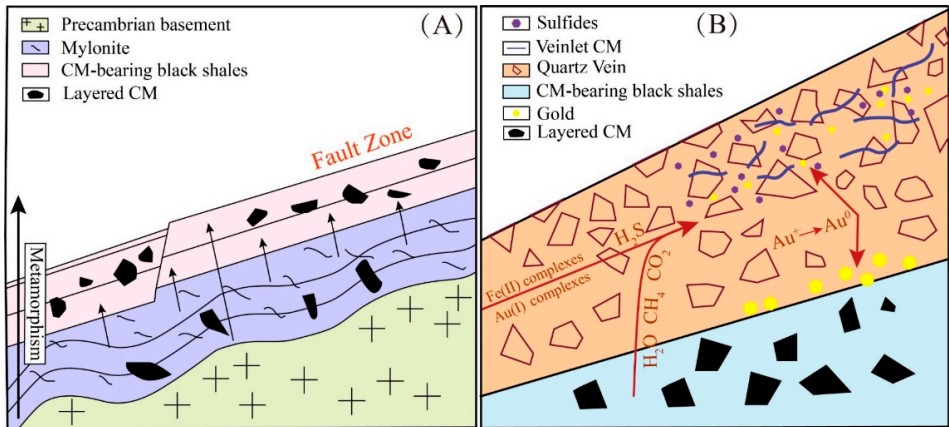

**Figure 9.** (**A**) CM-bearing black shales formed during metamorphism and early diagenesis in the Precambrian basement; (**B**) Schematic diagram showing the evolution of two types of CM in the Bumo deposit. (modified after KIRILOVA et al. 2018 and Wu et al. 2020 [51,59]).

## 7. Conclusions

(1) Two types of CM occur in the Bumo gold deposit, i.e., layered and veinlet CMs. Layered CM has a laminate structure, was formed at 450–550 °C and sourced from organic matter, and through pre-ore Caledonian metamorphism. In contrast, veinlet CM has the isometric shape and was formed at 200–350 °C, and through syn-ore hydrothermalism.

(2) Layered CM promoted the shear deformation of host rocks and provided a reducing environment, favorable for gold mineralization. Veinlet CM coprecipitated with sulfides and native gold.

**Supplementary Materials:** The following are available online at http://www.mdpi.com/2075-163X/10/8/708/s1. Table S1: Raman spectroscopy data of layered and veinlet CM samples from the Bumo deposit.

**Author Contributions:** Z.D., T.D., D.X., Z.L., and S.Z. designed the project; Z.D. did the original literature reviews; Z.D., Y.H., L.L., W.M., and K.X. performed the analyses, data collection, and data analysis; Z.D. and T.D. wrote and organized the paper, with a careful discussion and revision by Z.L., S.Z., and D.X. All authors have read and agreed to the published version of the manuscript.

**Funding:** The paper was financially co-supported by the National Natural Science Foundation of China (Nos. 41930428, 41472171, 41672077, 41302049), the DREAM project of MOST China (No. 2016YFC0600401), the Chinese Ministry of Land and Resources (No. 200646092), the Open Research Fund Program of Key Laboratory of Metallogenic Prediction of Nonferrous Metals and Geological Environment Monitoring (Central South University), Ministry of Education (Nos. 2019YSJS02, 2019YSJS12), Open Research Fund Program of State Key Laboratory of Nuclear Resources and Environment, East China University of Technology (No. NRE1915), and Jiangxi province graduate student innovation special fund project (YC2019-S271).

**Acknowledgments:** We are thankful for the assistance from reviewers. A particular thanks is given to the Hainan Geological Survey Institute and Zhaojin mining company for their help in the field investigation.

**Conflicts of Interest:** The authors declare no conflict of interest.

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
