# Peer review of "Genesis of Two Types of Carbonaceous Material Associated with Gold Mineralization in the Bumo Deposit, Hainan Province, South China"

_minerals, doi:10.3390/min10080708_

Round 1
Reviewer 1 Report
Add the specified references and correct English grammar and usage throughout manuscript.
- The authors of this manuscript claim that they have identified two types of carbonaceous material (CM) in a lode-gold deposit on Hainan Island. I agree with them, there definitely are two types of CM in their samples
- However, the first type of CM they studied is surely nothing more unique than metamorphosed carbonaceous black shale –the rock photos in Fig. 4a-b make this evident, along with their schematic diagram in Fig. 9
- Overall, the manuscript is fairly well-written. Of course, there are corrections to the grammar that can be made (e.g., Line 246, “There is two notable difference” should read “There are two notable differences”; many other errors of this kind exist in the manuscript – the authors need to spend time going through the whole paper to find and fix them), but I’ve seen worse.
- Lastly, there are some missing references which I have requested the authors add to their bibliography, and there are some sentences that need supporting references (indicated in the attached PDF).

Reviewer 2 Report
Dear Authors,
Please find my comments below and improve accordingly.
1. In abstract and conclusion, you believed that Au is present as bisulfide, however you do not have any evidence at this deposit. How can you assume? You have formation temperature information from previous research, but that is not enough to decide which gold complex is dominant in the responsible ore-forming fluid.
- What is a rectangle shown in Figure (B)?
- Do you have any subsurface geology information for Fig.2(B)? It is better to be shown.
- Your quartz in Figure 3(C) has a color, implying the sample is too thick for petrography. In addition, you should mention either plane polarized light (PPL) or cross polarized light (XPL)
- In Fig.4 caption, Au is NOT mineral name. You should improve and this au bearing mineral is one of the key to estimate hydrothermal activity for the formation of quartz vein hosting stringer CM.
- Hydrochloric acid is NOT HCL, but HCl. Please check it.
- In Section 4.1, there is repetitive expression. Please check and improve. In addition, you need to put the machine name for SEM and XRD in Section 4. And usually we do not use gold coating to observe gold bearing geological samples. Why did you use?
- In Fig. 5, please put sample ID in the caption. And, some of the units in X or Y axis were missing. Please check all the figures for the unit.
- In Table 1, you have two types of raw data, band peak /cm-1 and band area /cm-1. For the former, do you indicate peak position? How about the latter? Anyway, it is too far information to be able to understand the example shown in Figure 5.
- In Fig. 7, I could see Qtz peak around 20 degree as 2 theta, but not recognize quartz strongest peak at around 26.6 degree as 2 theta. Definitely, the peak is NOT at 25.6 degree as you mentioned in Fig.7 caption according to your graph. I am afraid your CM peak is for quartz.
- In Fig. 8, what is the meaning of the Y-axis variation? If none, you should use the bar to show the range for each locality as usually we did so.
- It is a bit related with my first question, but you did not indicate any movement of gold in Fig. 9. Why? You explained two scenarios for gold mineralization. One is at boundary between layered CM and quartz vein, and the other is in quartz vein with an association of stringer CM. Please explain again both discussion and conclusion to follow the scenarios by using Fig.9 appropriately.
Round 2
Reviewer 2 Report
*On previous comment 5, I am afraid you misunderstood my point.
Gold is element name, so you should use Native gold for mineral name.
*On previous comment 10, the two peaks for CM and quartz you identified seem quartz. Then, no more one peak for quartz as you pointed out at answer for my comment 10. Otherwise, why are you able to judge the presence of CM (graphite) by only one peak?
I suggest you that even overlapping of quartz and graphite at 26.5° is acceptable for your discussion (you have other evidence of the presence of CM).
*I noticed "FeO" in Fig 9 (b). Is it such a oxidative condition? or any other better expression such as Fe(II) or Fe(III)?
Thanks for your consideration.
Author Response
Please see the attachment.
Thanks again for your attention and kindly advice!
